# Prevalence of Abnormalities at Tandem Endoscopy in Patients Referred for Colorectal Cancer Screening/Surveillance Colonoscopy

**DOI:** 10.3390/cancers16233998

**Published:** 2024-11-29

**Authors:** George Triadafilopoulos

**Affiliations:** Department of Gastroenterology, Hepatology and Nutrition, University of Texas MD Anderson Cancer Center, 1515 Holcombe Blvd, Houston, TX 77030, USA; gtriadafilopoulo@mdanderson.org

**Keywords:** Screening for Barrett’s esophagus, screening for gastric cancer, endoscopy, colonoscopy, intestinal metaplasia, dysplasia

## Abstract

This retrospective analysis of selected patients who underwent guideline-based, point of care, endoscopy in tandem with guideline-based, screening/surveillance colonoscopy reveals potentially significant foregut pathology (not necessarily precancerous). Potentially precancerous foregut pathology is as prevalent as precancerous colon polyps found at colonoscopy. Benign foregut endoscopic and histologic findings may be amenable to dietary and pharmacologic interventions. Tandem EGD combines Barrett’s esophagus and gastric cancer screening in one examination and should be considered in all subjects referred for screening or surveillance colonoscopy.

## 1. Introduction

Colonoscopy, the most common screening test for colorectal cancer (CRC), is associated with reduced CRC incidence and mortality through early detection and treatment of cancer [1]. Adenomas are found in nearly 40% of screening colonoscopies in the US, and after their removal, patients undergo future surveillance. Several genetic polyposis syndromes are associated with an increased risk of CRC or cancers elsewhere in the GI tract, calling for the use of endoscopy for screening and surveillance of gastric and or duodenal abnormalities [2,3,4]. Performing elective upper (EGD) and lower (Colonoscopy) endoscopic procedures on the same day could be a patient-centered and less costly approach than a two-stage approach performed on different days, when clinically appropriate.

Recent guidelines [5] and epidemiologic reviews [6,7] suggest a single screening endoscopy for patients with chronic GERD symptoms, and three or more additional risk factors for Barrett’s Esophagus (BE) [8]. A consensus statement recommends *Helicobacter pylori* (HP) testing and endoscopic screening with biopsies should be offered at age 50 to those with a family history of gastric cancer (GC) in a first-degree relative, first-generation immigrants from high GC-incidence regions, and those belonging to racial/ethnic groups at increased risk [9]. A recent guideline suggests that patients 60 years or older with dyspepsia be investigated with EGD [10,11].

Performing a tandem EGD and colonoscopy in selected individuals has advantages, such as the following: it allows for the early detection of benign and/or precancerous foregut diseases; it is covered by insurance, as long as it is based on national guidelines; it decreases the endoscopists’ time demand, and improves patient transportation logistics; it is well-tolerated; it potentially reduces anesthesia risks; it improves efficiencies (scheduling and endoscopy unit utilization); and it provides the benefit of added therapies (diet, proton-pump inhibitor use, and endoscopic or surgical treatments). However, this approach may have disadvantages, such as the following: there is an unknown prevalence of these conditions in the general US population, despite its increasing diversity; its findings may generate anxiety produced by false-positive screening; it may lead to harm associated with further diagnostic testing; and it may lead to overdiagnosis of conditions that may be treated but would never have become clinically apparent. Finally, it has unproven cost-effectiveness.

In this study, we aimed to examine the prevalence of foregut endoscopic and histologic abnormalities in patients undergoing endoscopy, based on national guidelines, in tandem with point-of-care screening/surveillance colonoscopy for CRC. For this, we followed a rigorous endoscopic protocol, involving white-light and NBI chromoendoscopy in conjunction with meticulous histologic assessment using the Sydney classification, the Seattle biopsy protocol, and histologic assessment of the squamocolumnar junction (SCJ) and the duodenum in all patients. We wanted to (1) assess the implications for early cancer detection and intervention, and surveillance of potentially precancerous abnormalities (i.e., intestinal metaplasia and dysplasia), (2) identify benign foregut lesions that would benefit from intervention (i.e., peptic pathology, celiac disease), and, ultimately, (3) generate pilot data on the utilities of the tandem approach in such select patients.

## 2. Patients and Methods

### 2.1. Patients

This retrospective cohort study of prospectively collected information was approved by the MD Anderson Institutional Review Board. No individual study consents were obtained, and all data were kept confidential in an encrypted database. Our population consisted of consecutive subjects who underwent screening or surveillance colonoscopy in our ambulatory surgery center by the author, from 1 February 2023 to 31 May 2024. All patients were referred for CRC screening or surveillance (not diagnostic) colonoscopy—as an absolute inclusion criterion—to the outpatient MD Anderson Endoscopy facility in Woodlands, TX, USA. Patients did not volunteer foregut symptoms, such as dyspepsia or acid reflux, but they potentially reported such symptoms upon questioning. Based on their responses to prompting questions, and their personal or family history, or other risk factors, the screening nurse assigned them an indication score and added an EGD, to be performed in tandem, under one anesthesia. Before proceeding with the procedures, informed consent outlining the risks, benefits, alternatives, and potential complications was obtained and documented. Inclusion criteria were mainly based on national guidelines for BE (score > 4) and GC screening (score > 1) (Table 1), or other conditions justifying EGD in tandem with colonoscopy (such as Lynch syndrome, family history of esophageal or gastric cancer, or alarm symptoms, such as dysphagia or early satiety, if elicited. The breakdown of the justification for EGD in tandem with colonoscopy, the potential ICD-10 codes used for insurance authorization, and number of patients enrolled by indications are listed in Table 2. There were no other exclusion criteria. Specifically, subjects with prior gastrointestinal surgery for benign (i.e., bariatric) or malignant (i.e., partial colectomy for cancer) disease, or medication use (i.e., proton-pump inhibitors, aspirin or NSAID use) were not excluded.

During the study period, 1004 subjects underwent screening or surveillance colonoscopy by a single experienced endoscopist, using a standardized endoscopy and colonoscopy protocol, as described in detail in Methods (Appendix B). Of these, 317 subjects (32%) received tandem EGD and colonoscopy after having fulfilled the entry criteria justifying the EGD, and 688 only received colonoscopy. These two groups were then compared, based on the colonoscopy finding, to ensure that the dual procedure subjects were similar in terms of demographics, clinical indications for colonoscopy referrals, polyp burden, and related histology. Figure 1 depicts a flow diagram of the study populations and highlights clinical indications in each group.

### 2.2. Methods

After insurance clearance, patients received standard preparation for endoscopy and colonoscopy per MD Anderson guidelines, as needed, to reduce aspiration or bleeding risks. Specific procedural details are described in Appendix B.

## 3. Results

### 3.1. Patients and Overall Findings

Of the 317 subjects in the cohort who underwent tandem EGD and colonoscopy, there were 214 women (67.5%) and 103 men (32.5%). There were 237 Whites (74.7%), 16 Asians (5%), 40 Blacks (12.6%), and 24 Hispanics (7.5%). Their median age was 59 (age range 19–85). In contrast, of the 687 subjects undergoing only colonoscopy, there were 378 women (55%) and 308 men (45%). In this group, there were 520 Whites (75.6), 21 Asians (3%), 70 Blacks (10.1%), 68 Hispanics (9.9%) and 8 Other (1.2%). Their median age was 60 (age range 21–90). When comparing the colonoscopy findings between the two groups, the tandem EGD/colonoscopy one (*n* = 317) revealed 39 hyperplastic polyps (12.3%), 10 serrated polyps (3.1%), 118 tubular and tubulovillous adenomas (37.2%), and 2 cancers (1 colon and 1 rectal; 0.6%), while the colonoscopy-only group (*n* = 687) revealed 100 hyperplastic polyps (14%), 39 serrated polyps (5.7%), 338 tubular and tubulovillous adenomas (49.2%) and 10 colorectal cancers (1.4%). This analysis suggested that the colonoscopy was of superior quality in both groups, without any negative impact of the EGD on the colonoscopy polyp yield [12]. Figure 2 highlights the actionable findings on tandem EGD, and the actions taken at the discretion of the investigator.

Of the 317 subjects who underwent tandem EGD and colonoscopy, 6 (1.9%) had evidence of prior colocolonic or colorectal anastomosis, while 10 (3.5%) had prior ileocolonic anastomosis. Only two cases were incomplete (aborted) examinations, due to sigmoid strictures. Of the 687 subjects undergoing only colonoscopy, 46 (6.7%) had prior colocolonic, or colorectal anastomosis, while 26 (3.8%) had prior ileocolonic anastomosis. There were four subjects (4.6%) who had prior ileocolonic and colocolonic anastomosis, and one had colonoscopy through sigmoid colostomy. Only one case was an incomplete (aborted) examination, due to sigmoid stricture. At EGD, 20/317 patients (6.3%) revealed postoperative changes; as a result, select protocol biopsies could not be obtained. Ten had prior sleeve gastrectomy; eight had prior gastrectomy with Roux-en-Y anastomosis, one had prior gastrectomy with Billroth II anastomosis and prior esophagogastric anastomosis for proximal benign gastric tumor.

### 3.2. Endoscopy and Biopsies

Duodenum: Per-protocol endoscopic assessment of the duodenum revealed peptic duodenitis in 23 patients; only 6 were associated with histologic erosive duodenitis. One patient had an *H. pylori*-negative bulbar ulcer. Two patients exhibited post-bulbar duodenal strictures (all dilated using the endoscope). In total, 26 patients (8.2%) were classified as peptic duodenitis. All these patients were treated with oral omeprazole, with 20 mg daily for 2 months. The remaining had normal duodenal appearance. Protocol bulbar and post-bulbar biopsies of the duodenum revealed 18 cases of duodenal lymphocytosis, 3 of which had associated villous blunting, consistent with celiac disease (CD). All were treated with gluten-free diet (Figure 3 and Figure 4).

Pylorus: Six patients (1.9%) were noted to have subtle benign pyloric stenosis, which was dilated using the endoscope, followed by treatment using oral omeprazole 20 mg daily for 2 months. No distinct pyloric ulcer was seen; no biopsies were obtained from the pylorus.

Antrum: Endoscopically, the antrum showed non-specific gastritis in 45/317 patients (14.1%). Of those, 14 were infected with *H. pylori* and they were all treated with triple therapy. Antral nodularity was seen in five patients (1.6%). There was no relationship between *H. pylori* infection and visible gastritis. Random (protocol) antral biopsies revealed gastric intestinal metaplasia GIM (one with dysplasia and one related to *H. pylori*) in 20 subjects (Figure 3 and Figure 4).

Incisura: Endoscopically, the incisura showed non-specific gastritis in 42/317 patients (13.2%). Of those, 14 were infected with *H. pylori* and they were all treated with triple therapy (the same as above). There was no relationship between *H. pylori* infection and visible gastritis. Random (protocol) incisura biopsies revealed GIM (one with dysplasia, and one related to *H. pylori*) in seven subjects. In addition, select incisura biopsies revealed fibrosis (*n* = 3), hemorrhage (*n* = 1) and autoimmune metaplastic atrophic gastritis (AMAG) (*n* = 3) (Figure 3).

Gastric body: Endoscopically, the gastric body showed non-specific gastritis in 20/317 patients (6.3%). Of those, 17 (5.3%) were infected with *H. pylori* and they were all treated with triple therapy. There was no relationship between *H. pylori* infection and visible gastritis. Random (protocol) gastric body biopsies revealed GIM (one with dysplasia, and one related to *H. pylori*) in 14 subjects (Figure 4). Select biopsies revealed fibrosis (*n* = 3), hemorrhage (*n* = 1), atrophy (*n* = 2), neuroendocrine cell hyperplasia (*n* = 1), and autoimmune metaplastic atrophic gastritis (AMAG) (*n* = 6). Fundic polyps (FPs) were seen and confirmed benign histologically in 31 cases (9.8%) (Figure 4).

Cardia: Endoscopically, the cardia revealed infiltrative tumor in three patients (two gastric, and one esophageal; both were confirmed histologically). It was normal in 314 cases. Protocol biopsies were obtained universally 0.5 cm below the SCJ. In these, 129 (41%) showed chronic inflammation (12 with histologic *H. pylori* infection); GIM was found in 14 (4.4%) of subjects (none related to *H. pylori* infection). Parietal cell hyperplasia was noted in nine cases, and atrophy and fibrosis in three. Pancreatic acinar metaplasia (PAM) was seen in six cases (1.9%). Sliding hiatal hernia was noted in 103/317 subjects (32.5%) and was associated with Schatzki’s ring in 4 patients. Overall, 11 subjects (3.5%) exhibited Schatzki’s ring; one required dilation (Figure 3 and Figure 4).

Squamocolumnar junction: Endoscopically, the SCJ showed two cancers (one gastric and one esophageal, confirmed histologically, see above) and peptic esophagitis extending upwards to the distal esophagus in 25/317 subjects (8%).

Twenty-four patients had erosive esophagitis in four quadrant biopsies of the SCJ (7.6%) and two had histologic ulcers. All were treated with oral omeprazole, with 20 mg daily for 2 months. Fifty-six subjects were found to have IM at the SCJ (17.6%), variably associated with GIM elsewhere (see below); four had LGD and one HGD, and were referred for endoscopic eradication therapy. There were 21 cases of PAM (6.6%), 3 of them occurring in conjunction with IM (but no dysplasia); if cardiac PAM was added, which was seen in six cases (1.9%), the total % prevalence of PAM in that region is 8.5% (Figure 3 and Figure 4).

Esophagus: Endoscopically, the tubular esophagus revealed peptic esophagitis in 25/317 subjects (8%). These were distributed as the following: 13 LA-Grade A; 15 LA-Grade B; 2 LA-Grade C and 5 LA-Grade D. There were seven patients with endoscopically visible Barrett’s esophagus (BE), five containing dysplasia. All were treated with oral omeprazole, with 20 mg daily for 2 months. Histologically, there were eight cases of eosinophilic esophagitis. There were nine patients with histologic erosive esophagitis on distal esophageal biopsies and 26 patients had histologic changes of GERD with papillary hyperplasia and chronic inflammation without erosions or ulcers, totaling 35 (11%) of patients (Figure 3 and Figure 4). All were treated using daily omeprazole, with 20 mg for 2 months. One patient had esophageal *Candidiasis* and was treated with fluconazole 100 mg orally for 10 days.

Combination findings: Appendix A depicts a diagram of biopsy protocol at endoscopy. In addition to station biopsies, focal, endoscopic abnormalities were separately biopsied or resected. Immunohistochemistry for *H. pylori* infection was obtained in all cases. Based on the focal (isolated), or contiguous presence of GIM of the antrum, incisura, gastric body, and cardia, as well as IM of the SCJ, 86/317 subjects (27%) were recommended endoscopic surveillance, 3 years after endoscopy (Appendix A). In addition, 5/86 cases contained low- or high-grade dysplasia at SCJ and were referred for endoscopic eradication therapy (EET) (Figure 2).

## 4. Discussion

In this study of select patients referred for screening or surveillance colonoscopy who also underwent a tandem EGD (32% of total cohort), we identified actionable benign (45%), premalignant (31%), and malignant (0.9%) findings that are comparable to the precancerous polyp (40.3%) and malignancy (0.6%) yield of colonoscopy. Tandem endoscopy combines BE and GC screening in one examination and is practically and fiscally attractive, with the potential for saving and improving lives. Thus far, screening for BE is considered separately in Western, mostly Caucasian, populations with GERD, and screening for GC is considered separately in Asian and Hispanic populations with high incidence of *H. pylori* infection, without considering both concomitantly, thereby potentially maximizing the cost-effectiveness of future surveillance. Our findings validate the national guidelines [5,6,7,8] recommending screening for EGD, and they provide additional justification by identifying actionable benign endoscopic and histologic abnormalities. Further, substantiated by rigorous and hitherto never-implemented histologic assessment, our study provides valuable histologic information on the prevalence of precancerous abnormalities, such as intestinal metaplasia, with resulting implications for cancer screening and surveillance.

Same-day upper and lower endoscopy has many potential advantages, by reducing costs, shortening hospital stay, and expediting patient care, but its yield and cost-effectiveness remain unclear [13]. The benefits of performing an endoscopy must be balanced against its potential harms, and such practice should follow society guidelines [14]. Inappropriate use of upper endoscopy exposes patients to unnecessary risks and financial burden. A meta-analysis of 53,392 patients identified a high frequency of inappropriate indications for upper endoscopy and a higher diagnostic yield in patients who had an appropriate indication [15]. Physician offices and ambulatory surgery centers have much higher different-day procedure rates compared with hospital outpatient departments [16], representing an opportunity for quality improvement and financial savings. The sequence (upper–lower) or lower–upper of same-day double gastrointestinal endoscopy does not affect total procedure time or medication use [17].

GERD symptoms are an essential criterion for BE and EAC screening, but they are not universally adopted. A systematic review and meta-analysis examined the prevalence of BE/EAC in those with and without GERD [18]. The overall risks for BE and long-segment BE are higher in patients with GERD, but the risk for short-segment BE did not differ between the two groups. In nine population-based studies (2244 patients with and 3724 patients without GERD), BE prevalence in patients without GERD was 4.9% (95% CI, 2.6–9%). BE prevalence was highest in North American studies (10.6% [GERD] and 4.8% [non-GERD]). Given the substantial BE prevalence in those without GERD, future BE/EAC early-detection guidelines may need revision.

It is important to examine to what degree our endoscopic and histologic findings were clinically significant (actionable), innocuous, or harmful. Duodenal lymphocytosis (DL), noted in 5.7% of our cohort, is a non-specific histologic alteration of unknown cause, at times associated with dyspepsia, as it was in 17% of our cases. DL corresponds to Marsh grade 1, that is, mild inflammation of the duodenal mucosa, with no crypt hyperplasia or villous atrophy. About 11% of DL patients exhibit clinical and serologic features of gluten sensitivity. In our cohort, three subjects had villous blunting, suggestive of CD. *Helicobacter pylori* gastritis may also contribute to DL, which may resolve following eradication, but it was not seen in our patients [19]. In all our cases of DL, a gluten-free diet was implemented (Figure 3).

Active *Helicobacter pylori* infection, seen in 6% of our cohort (Figure 3), denotes a risk of chronic gastritis, peptic ulcer disease, and gastric neoplasia. Because most infections remain clinically silent, *H. pylori* case-finding thus far is based on symptoms (i.e., dyspepsia), or high-risk indicators, such as racial or ethnic background and family history, like in our cohort. However, this approach misses many infected individuals who remain at risk of GC, hence our use of the Sydney protocol. Individuals with chronic *H. pylori* are at risk of gastric preneoplasia, which is also asymptomatic and only reliably diagnosed using an endoscopy and biopsy. Thus, to make a significant impact on gastric cancer prevention, a systematic approach is needed. *H. pylori* eradication must also be optimized, given sharply decreasing rates of success with common therapies and increasing antimicrobial resistance [20]. All our identified *H. pylori* subjects were treated with triple therapy and were confirmed eradicated by stool antigen testing 2–3 months after therapy, without any evidence of antimicrobial resistance. Those with gastric pre-neoplasia—HP-related or not—were then entered into endoscopic surveillance.

Gastric intestinal metaplasia (GIM), be it of samples of endoscopic abnormalities (i.e., nodules) or randomly, using the Sydney protocol, was noted in 40 (26%) of our cohort (Figure 4 and Appendix A). Because of its risk for gastric neoplasia, particular attention was paid to identify the concomitant *H. pylori* infection and treat it, but also to exclude dysplasia. Regardless, all patients with GIM were entered into endoscopic surveillance in 3-year intervals (Figure 3)**.** AMAG, found in 10 patients (3.1%) of our cohort, is an inherited autoimmune disease and has a 3-fold risk of gastric adenocarcinoma. A surveillance endoscopy every 3 years should be considered in individuals with atrophic gastritis, based on anatomic extent and histologic grade, as it was done in all our cases [21].

Fundic polyps (FPs) are common (9.8% in our cohort) and benign, with little malignant potential. Their prevalence has been increasing with the widespread and frequent use of PPIs. Most FPs occur in the gastric body, are small (<5 mm) and sessile, and surveillance is not required. However, histopathology is necessary if endoscopic findings different from ordinary FPs are noted [22]. In all our cases, representative FPs were biopsied to confirm histology; there were no polyps in our cohort with disturbing endoscopic characteristics or size that required polypectomy.

Careful visualization of the cardia in antegrade and retrograde views was universally performed and combined with endoscopic assessment for hiatal herniation. Universally, biopsies were obtained 0.5 cm below the squamocolumnar junction (SCJ) to assess for GIM and its possible link to GIM of the stomach, gastric atrophy, and SCJ-IM. Intestinal metaplasia in the esophagus (Barrett’s esophagus IM, or BE-IM) and stomach (GIM) are considered precursors for esophageal and gastric adenocarcinoma, respectively. Studies have revealed the gradual molecular and phenotypic transition from a gastric to intestinal phenotype (IM) in the esophagus and stomach. Because BE-IM and GIM can predispose to cancer, this new understanding of a common developmental trajectory leads to a more unified approach to detection and treatment [23]. Indeed, this concept served as an initial thrust to this clinical study, namely, the assessment of prevalence and further phenotypic characterization of these regions during one endoscopic procedure. Although short-segment gastric metaplasia (SS-GM) is a distinct entity from short-segment intestinal metaplasia (SS-IM) and the malignant potential of GM is lower than IM, a thorough pathology and endoscopy review has questioned whether SS-GM warrants inclusion in BE surveillance [24]. We did not encounter any cases of SS-GM in our cohort.

SCJ-IM is a common finding in GERD, and it may represent an early step in the development of esophagogastric junction adenocarcinoma (EGJAC) in the West. In our cohort, 17.6% exhibited SCJ-IM, some 2.2% associated with short-segment BE, others with GIM of the cardia (1.6%) or the stomach. Worldwide, SCJ-IM may represent progression along the Correa cascade triggered by *Helicobacter pylori*. Using decision analytic modeling that evaluated the cost-effectiveness of endoscopic surveillance of EGJ-IM, low-intensity longitudinal surveillance (every 5 years) was cost-effective in populations with higher EGJAC incidence [25]. Aiming at estimating lifetime benefits, complications, and cost-effectiveness of GIM surveillance using EGD, another study concluded that surveillance of incidentally detected GIM every 5 years is cost-effective [26]. These models unfortunately address distinct but overlapping issues, those of EAC, EGJAC and GC, without considering EGD as the sole tool for clinical exploration. It is our hope that, particularly in the modern era of mass migration and globalization, this pilot study will initiate further real-world studies evaluating the impact of screening/surveillance on the combined incidence and mortality of foregut cancers and their ominous implications. Indeed, in our limited cohort, three EGJACs were noted, only one reporting dysphagia and two reporting dyspepsia, upon questioning prior to colonoscopy.

PAM and IM overlap infrequently at the gastroesophageal junction/distal esophagus (GEJ/DE). A cohort study suggested that PAM at the GEJ/DE is associated with protective effect against IM [27]. In our cohort, 8.2% were noted to have PAM of the cardia (*n* = 6) and at the SCJ (*n* = 20) alone, or in conjunction with IM (*n* = 3) but inadequate to make any claims of clinical importance. Nevertheless, in the context of larger prospective assessments, particularly using immunohistochemical studies for p53, the role of PAM will be further clarified.

Given the low prevalence of *H. pylori* in our cohort, we suspect that aspirin and NSAIDs, used by approximately 11% of the U.S. population [28,29], contributed to peptic lesions and the occurrence of GERD symptoms and dyspepsia. Overall, tandem EGD identified and allowed PPI treatment in 37% of patients, comprising benign peptic (gastro-duodenal and esophageal) lesions, both for symptoms and prevention of complications. In addition, anti-HP therapy was given in 18 patients (6%), while endoscopic dilation was applied in 10 patients (3%) (Figure 3).

Figure 2, Figure 3 and Figure 4 highlight the benefits of the tandem approach. Collectively, the identification of actionable benign (45%) and malignant or premalignant (32%) lesions at EGD, matched those with actionable benign (12%) and premalignant polyps (40.3%) and CRC (0.6%) at colonoscopy. We believe that the yield of important pathology in our cohort—carefully selected on meeting screening criteria for BE, GC, or both—together with history of inherited polyposis syndromes, family history of esophageal and gastric cancer, and dysphagia and early satiety, amply justify the tandem approach. We did not identify any actionable foregut pathology in our 18 patients with inherited polyposis syndromes.

Our dual same-day procedure approach has a multidimensional impact on institutional resources, due to the added endoscopy personnel, and facility utilization, increasing by 15–30 min per patient. The fiscal impact, through identification of foregut pathology, will generate billing from surgical pathology, additional billing for physician services and downstream billing from imaging, including EUS, or downstream billing from surgical and oncology services. All these will need to be considered and justified, prior to a general implementation of this approach to ensure cost-effectiveness, but it is beyond the scope of our pilot effort. There is also a research impact: based on preliminary data, we could examine the role of additional screening modalities, currently approved (i.e., TissueCypher), participate in trials of other new screening modalities (i.e., liquid biopsies), and possibly capitalize on meticulously collected biopsies acquired for basic research endeavors on disease genetics, proteomics, etc., using microdissection [30].

There are limitations in this study that could be addressed prospectively, based on formally collected clinical research forms (CRFs) by uniformly trained and operating clinical coordinators, more accurately soliciting information, not only based on the BE and GC screening criteria (or both), but also on standardized semiquantitative symptom questionnaires, more elaborate family history information, and on a larger and more racially and ethnically diverse patient population presenting for colonoscopy to multiple endoscopists, with variable expertise in recognition and endoscopic sampling and removal of both foregut and colonic pathology. The data procured in this study were limited to an outpatient sample who were ASA class I-II, with insurance coverage, seen at a suburban outpatient endoscopy facility, and potentially not representing the larger metropolitan Houston area in diversity, attitude, and discipline towards cancer screening and surveillance. Further, a uniform histopathologic interpretation of precancerous and malignant lesions, preferably by two independent pathologists (plus a referee), enhanced by immunohistochemical and molecular analyses, would provide more precise insights. By integrating cutting-edge imaging technology with advanced software, digital pathology may improve quality assurance and standardization. Artificial intelligence significantly improves cancer diagnosis, classification, and prognosis. It also enhances the spatial analysis of the tumor microenvironment and enables the discovery of new biomarkers, advancing their translation for therapeutic applications [31]. Finally, actions taken in this cohort were solely based on one physician’s attitude, not representing a collective, protocol-driven clinical management of actionable findings; this could only be done in a prospective, multi-site, collaborative effort. Nevertheless, this is a real-life study providing intriguingly promising information on foregut/hindgut screening and surveillance, not heretofore properly addressed.

## 5. Conclusions

In conclusion, our analysis of selected patients who undergo guideline-based, point-of-care endoscopy, in tandem with screening or surveillance colonoscopy reveals significant foregut pathology (not necessarily precancerous), and merits formal prospective study. Our approach identifies potentially actionable foregut pathology that is almost as prevalent as precancerous colon polyps and cancer. Beyond further endoscopic surveillance for preneoplasia, benign foregut endoscopic and histologic findings may be amenable to other interventions. Yet, the long-term implications and cost-effectiveness are uncertain, and will require further study.

## Figures and Tables

**Figure 1 cancers-16-03998-f001:**
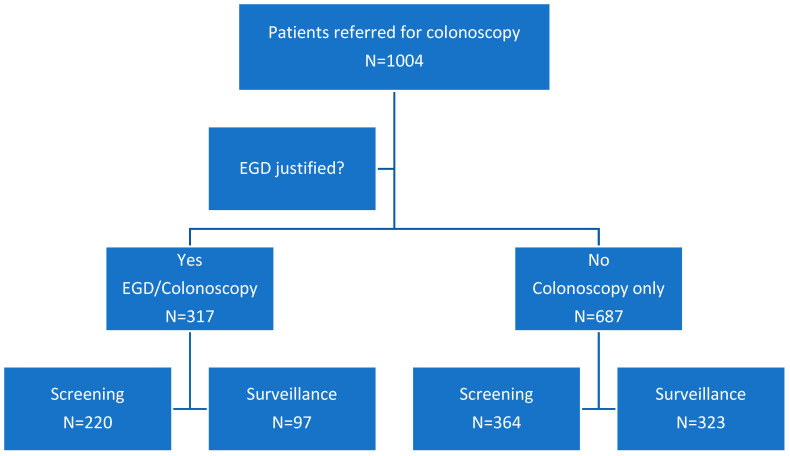
Study flow diagram.

**Figure 2 cancers-16-03998-f002:**
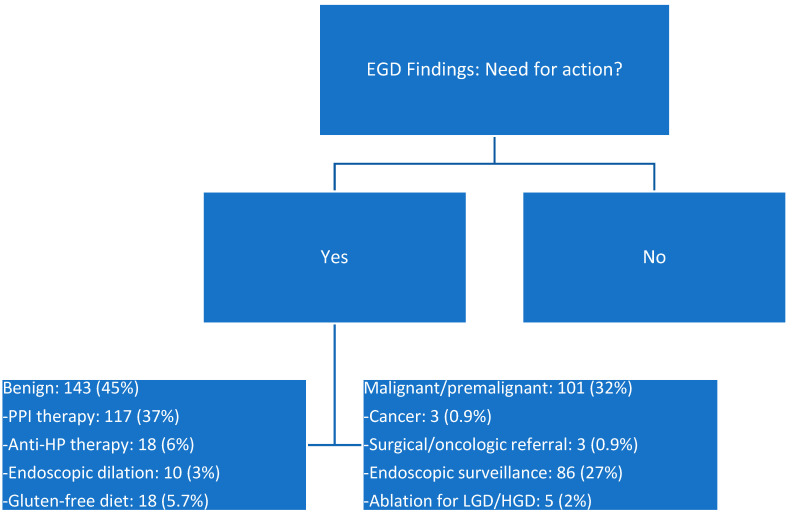
Need for action depending on endoscopic and histologic findings at endoscopy in the 317 subjects of the dual EGD/colonoscopy cohort.

**Figure 3 cancers-16-03998-f003:**
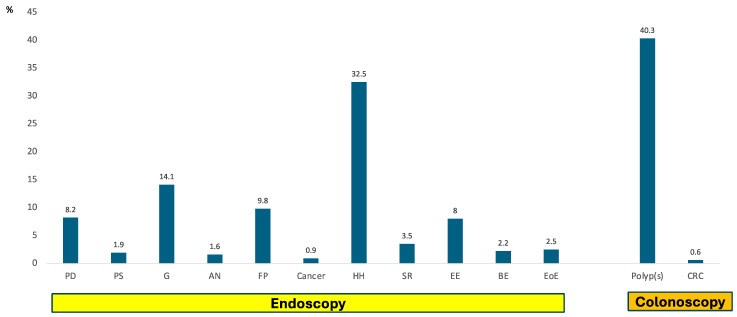
Point % prevalence of key benign and malignant endoscopic abnormalities at tandem EGD and colonoscopy in 317 subjects. PD: Peptic duodenitis; PS: Pyloric Stenosis; G: Gastritis; AN: Antral Nodule; FP: Fundic Polyp; Cancer; HH: Hiatal Hernia; SR: Schatzki’s Ring; EE: Erosive Esophagitis; BE: Barrett’s Esophagus; EoE: Eosinophilic Esophagitis; Polyps; CRC: Colorectal Cancer.

**Figure 4 cancers-16-03998-f004:**
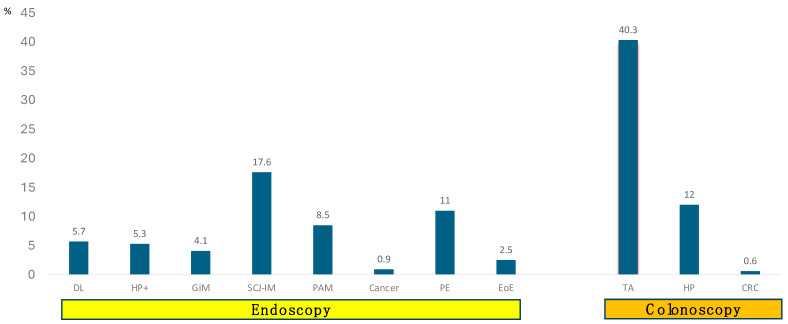
Point % prevalence of key histologic abnormalities at tandem EGD and colonoscopy in 317 subjects. DL: Duodenal lymphocytosis; HP+: Active *H. pylori* infection; GIM: Gastric Intestinal Metaplasia; SCJ-IM: Squamocolumnar Junction–Intestinal Metaplasia; PAM: Pancreatic Acinar Metaplasia; Cancer; PE: Peptic esophagitis; EoE: Eosinophilic esophagitis; TA: Tubular Adenoma (including serrated polyps); HP: Hyperplastic polyps; CRC: Colorectal cancer.

**Table 1 cancers-16-03998-t001:** Guideline-based criteria for BE and GC screening.

Screening Criteria for BE	Yes (1); No (0)	Screening Criteria for GC	Yes (1); No (0)
Chronic heartburn and regurgitation		First-degree relative with GC	
Male		First-generation immigrants from high *H. pylori*-prevalence regions	
Age > 50		African Americans, Alaskan Natives, American Indians, Asian Americans, and Hispanic Americans	
White		Age > 60 with dyspepsia *	
Tobacco			
Obesity (BMI > 30)			
**Total**	**EGD if score > 4**	**Total**	**EGD if score > 1**

* Dyspepsia: A persistent or recurrent pain, or discomfort in the upper abdomen.

**Table 2 cancers-16-03998-t002:** Justification for EGD in the study cohort.

Indication	Potential ICD-10 Codes Used	Number of Patients
Criteria for BE screening met (Score > 4)	Z13.810; K21.9; R13.10	132
Criteria for GC screening met (Score > 1)	Z13.810; K30	86
Combined BE and GC criteria	Z13.810; K30; K21.9	38
Inherited polyposis syndrome (Lynch, JPS, FAP)	D13.91; Z15.09; D12.6	18
FH esophageal or gastric cancer	Z80.0	25
Miscellaneous alarm symptoms (dysphagia, early satiety)	R13.10; R68.81	18

## Data Availability

The original contributions presented in the study are included in the article/Appendix A, further inquiries can be directed to the corresponding authors.

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
