# Peer review of "Prevalence of Abnormalities at Tandem Endoscopy in Patients Referred for Colorectal Cancer Screening/Surveillance Colonoscopy"

_cancers, 2024, doi:10.3390/cancers16233998_

Round 1

Reviewer 1 Report

Comments and Suggestions for Authors

Overall a nice paper, well-written (despite quite verbose) regarding an approach to identify new gastric cancer utilizing EGD when doing colonoscopy in high-risk patients.

1. Currently the paper is framed for gastroenterology, not oncology. Please move all non-cancer related conditions to supplements. Please focus the analysis to H.pylori, pre-cancer lesions, and gastric cancers.

2. Is intestinal metaplasia in stomach really pre-cancer? Can you please limit the analysis to true pre-malignant and malignant lesions? Even if the % is small, I think it is still a strong sell to consider tandem EGD as part of screening in high risk populations.

3. Methods section in regards to EGD and colonoscopy (2.4 and 2.5) technique and pathology evaluation seem more appropriate for a gastroenterology journal, not appropriate for oncology journal. Please make these sections much shorter.

4.  figure 2 is excellent.

5. Again results section, there is too much description on benign lesions, that is not relevant for oncology journal.

6. Again discussion section, there is too much focus on pathology findings that is recommended for surveillance, really the focus should be finding gastric cancer in an oncology journal. The current manuscript is more appropriate for gastroenterology journal, but I think if the manuscript can be written in a more concise manner focusing on the oncology components, that would be most excellent! I think a lot of oncology readers would be interested.

Author Response

Overall, a nice paper, well-written (despite quite verbose) regarding an approach to identify new gastric cancer utilizing EGD when doing colonoscopy in high-risk patients.

I thank the reviewer for their time spent reviewing my paper and their generally favorable comments. To clarify, the proposed strategy is intended to identify not only gastric, but also esophageal precancer and cancer. Currently, we screen for Barrett’s esophagus or gastric precancer separately and, based on cancer statistics, ineffectively.

My apologies for the verbosity, which was intended to enhance clarity for a wider audience (see below).

Currently, the paper is framed for gastroenterology, not oncology. Please move all non-cancer related conditions to supplements. Please focus the analysis to H. pylori, pre-cancer lesions, and gastric cancers.

I respectfully disagree with the notion that the paper is framed for gastroenterology, not oncology. The purpose of our study was to examine the value of screening for foregut pathology in the setting of a screening colonoscopy. The ordering physicians in all our study patients were not gastroenterologists, but oncologists and cancer prevention professionals. These are the ones that need to learn about the merits of our tandem screening approach, since they are the ones who will be ordering the screening endoscopic procedures, to be done by gastroenterologists, as needed.

Further, the benign findings of our study warrant attention, again by the referring physicians, since they contribute to patients’ quality of life impairment and provide opportunity for improvement with therapy (i.e., acid suppression). The presence of such “benign findings” in turn may alter choices regarding chemotherapy or radiation in select patients.

  1. Is intestinal metaplasia in the stomach really pre-cancer? Can you please limit the analysis to true pre-malignant and malignant lesions? Even if the % is small, I think it is still a strong sell to consider tandem EGD as part of screening in high-risk populations.

Intestinal metaplasia of the stomach is considered precancerous (Sugano K, Moss SF, Kuipers EJ. Gastric Intestinal Metaplasia: Real Culprit or Innocent Bystander as a Precancerous Condition for Gastric Cancer? Gastroenterology. 2023 Dec;165(6):1352-1366.e1). Figures 3 and 4 clearly separate pre-malignant (GIM, SC-IM) and malignant (Cancer) lesions.

Because BE-IM and GIM can predispose to cancer, this new understanding of a common developmental trajectory leads to a more unified approach to detection and treatment (Nowicki-Osuch K, Zhuang L, Cheung TS, Black EL, Masqué-Soler N, Devonshire G, Redmond AM, Freeman A, di Pietro M, Pilonis N, Januszewicz W, O'Donovan M, Tavaré S, Shields JD, Fitzgerald RC. Single-Cell RNA Sequencing Unifies Developmental Programs of Esophageal and Gastric Intestinal Metaplasia. Cancer Discov. 2023 Jun 2;13(6):1346-1363; Black EL, Ococks E, Devonshire G, Ng AWT, O'Donovan M, Malhotra S, Tripathi M, Miremadi A, Freeman A, Coles H; Oesophageal Cancer Clinical and Molecular Stratification (OCCAMS) Consortium; Fitzgerald RC. Understanding the malignant potential of gastric metaplasia of the oesophagus and its relevance to Barrett's oesophagus surveillance: individual-level data analysis. Gut. 2024 Apr 5;73(5):729-740. Indeed, this concept served as an initial thrust to this clinical study, namely, the assessment of prevalence and further phenotypic characterization of these regions during one endoscopic procedure.

  1. Methods section regarding EGD and colonoscopy (2.4 and 2.5) technique and pathology evaluation seem more appropriate for a gastroenterology journal, not suitable for oncology journal. Please make these sections much shorter.

I agree. The methodological details, which are essential from both the endoscopic and histopathologic perspectives, have now been moved to the Appendix.

  1. Figure 2 is excellent.

Thank you

  1. Again results section, there is too much description on benign lesions, that is not relevant for oncology journal.

The benign findings of our study warrant attention, again by the referring physicians, since they contribute to patients’ quality of life impairment and provide opportunities for improvement with therapy (i.e., acid suppression). The presence of such “benign findings” in turn, may alter choices regarding chemotherapy or radiation in select patients.

  1. Again, discussion section, there is too much focus on pathology findings that is recommended for surveillance, really the focus should be finding gastric cancer in an oncology journal. The current manuscript is more appropriate for gastroenterology journal, but I think if the manuscript can be written in a more concise manner focusing on the oncology components, that would be most excellent! I think a lot of oncology readers would be interested.

We understand the reviewer’s reservations. Yet, we need to acknowledge that, if our proposed tandem approach is to be implemented, it will start with oncologists and cancer prevention professionals who will (1) refer their patients to gastroenterologists and (2) expect a thorough and standardized endoscopic/histologic foregut assessment. This is not currently done, hence the disturbingly unchanged epidemiology of these cancers. It is hoped that our approach will have an impact on foregut cancer prevention in the increasingly diverse US population.

Reviewer 2 Report

Comments and Suggestions for Authors

The authors presents his experience of simultaneous examination of upper and lower gastrointestinal tract upon screening. It is self-explanatory that subjects undergoing endoscopic examination of the upper gastrointestinal tract will have some rate of actionable findings, irrespective whether this procedure is performed at the same day with colonoscopy or not. Phrasing is confusing due to the use of professional jargon: “endoscopy” is not necessarily examination of the stomach, this group of methods can be used for examination of other organs. Many abbreviations are not explained upon the first mention. The paper is absolutely too long. The presentation of the data and discussion should focus on advantages and advantages of tandem examination of upper and lower parts of gastrointestinal tract. The descriptive information, e.g., characteristics of identified lesions, can be moved into supplement.    

Author Response

The author presents his experience of simultaneous examination of upper and lower gastrointestinal tract upon screening. It is self-explanatory that subjects undergoing endoscopic examination of the upper gastrointestinal tract will have some rate of actionable findings, irrespective whether this procedure is performed at the same day with colonoscopy or not. Phrasing is confusing due to the use of professional jargon: “endoscopy” is not necessarily examination of the stomach, this group of methods can be used for examination of other organs. Many abbreviations are not explained upon the first mention. The paper is absolutely too long. The presentation of the data and discussion should focus on advantages and advantages of tandem examination of upper and lower parts of gastrointestinal tract. The descriptive information, e.g., characteristics of identified lesions, can be moved into supplement.    

I agree. I tried to address the various comments of the reviewer, mostly shortening the paper and moving things in Appendix 1. The results section is detailed because it outlines the details of the histopathology and its relevant clinical implications

Reviewer 3 Report

Comments and Suggestions for Authors

This manuscript is an original article that retrospectively examined the prevalence of foregut endoscopic and histologic abnormalities in totally 317 subjects referred for screening/surveillance colonoscopy who also underwent a tandem endoscopy. The study showed that actionable benign (45%) and malignant/premalignant (32%) findings were identified in the tandem endoscopy.

This study was conducted well, and the methods are appropriate. The results will be of interest to clinicians in the field.

However, the following major and minor issues require clarification:

Major

1.     Tandem endoscopy is usually performed in the clinical setting. This study is valuable to evaluate the prevalence of upper gastrointestinal abnormal findings in a tandem endoscopy. However, these results obtained in this study is predictable to be similar to those in the whole population in U.S. Therefore, I recommend that the authors also examine and show the advantages of the tandem endoscopy, including cost, procedure time, tolerance, and safety (complications), compared to a separately- performed endoscopy.

Minor

1.     (P1L25) Cancer is included in premalignant findings.

2.     (P5L150) “Figure 1” should be replaced with “Supplemental Figure S1”.

3.     (P6L218) “CD” should be written with unabbreviated form.

4.     Interval with prior upper endoscopy should be provided as it can influence the results.

5.     Results seems somewhat redundant. Treatment can be deleted, and clinically- unimportant endoscopic findings can be omitted. Accordingly, Figure 3 and 4 should be modified or omitted.

Author Response

This manuscript is an original article that retrospectively examined the prevalence of foregut endoscopic and histologic abnormalities in 317 subjects referred for screening/surveillance colonoscopy who also underwent a tandem endoscopy. The study showed that actionable benign (45%) and malignant/premalignant (32%) findings were identified in the tandem endoscopy.

This study was conducted well, and the methods are appropriate. The results will be of interest to clinicians in the field.

I thank the reviewer for their overall favorable assessment. Specific edits are addressed below

However, the following major and minor issues require clarification:

Major

Tandem endoscopy is usually performed in the clinical setting. This study is valuable to evaluate the prevalence of upper gastrointestinal abnormal findings in a tandem endoscopy. However, these results obtained in this study are predictably similar to those in the whole population in U.S. Therefore, I recommend that the authors also examine and show the advantages of the tandem endoscopy, including cost, procedure time, tolerance, and safety (complications), compared to a separately- performed endoscopy. 

Tandem endoscopy is performed in the clinical setting only for dual indications, such as gastrointestinal bleeding of unknown origin. This study is conceptually different: Before a screening or surveillance colonoscopy, the requesting oncologist or cancer prevention professional interrogates about foregut symptoms or risk factors and justifies (based on US guidelines) the addition of a tandem screening endoscopy.  In our study, this was feasible only in 32% of cases. Such cases would have potentially been missed if subjects were conventionally referred for CRC screening.

Our results cannot be predictably similar to the US population due to subject selection justifying screening colonoscopy for CRC, and screening endoscopy for Barrett’s esophagus and gastric neoplasia.

We do not expect that a non-tandem endoscopy would yield differential clinical or fiscal results. Our study aimed to capitalize at capturing individuals referred for CRC screening who would qualify for tandem endoscopy, based on national guidelines. This is not currently practiced. Further, an upper endoscopy is done to screen for Barrett’s esophagus or gastric cancer, not both, in select subjects usually separately from CRC screening, which is typically streamlined directly.

Minor

  1. (P1L25) Cancer is included in premalignant findings.

Corrected, thank you

  1. (P5L150) “Figure 1” should be replaced with “Supplemental Figure S1”.

Corrected; thank you

  1. (P6L218) “CD” should be written with unabbreviated form.

Corrected; thank you

  1. Interval with prior upper endoscopy should be provided as it can influence the results.

Unfortunately. we do not have this information. This was a prevalence study of subjects who did not complain of foregut symptoms, but they responded positively to questions, thereby justifying the endoscopy. Some were under empirical treatment for symptoms (i.e., with acid suppressants for chronic reflux) without having had an endoscopy.

  1. Results seem somewhat redundant. Treatment can be deleted, and clinically- unimportant endoscopic findings can be omitted. Accordingly, Figure 3 and 4 should be modified or omitted.

This study describes heretofore unrecognized endoscopic and histopathologic findings that carry implications for improved foregut cancer screening. The details provided herein are important in designing the proper cost-effectiveness screening models, aiming at combining screening for esophageal and gastri neoplasia. Hence, they are important and stated in the results while summarized in Figures 3 and 4.

Round 2

Reviewer 2 Report

Comments and Suggestions for Authors

Some suggestions have been addressed with proper diligence, while some have not

Author Response

I thank the reviewer for their reply. They have no specific further requests and they appear satisfied with our first revision.

Reviewer 3 Report

Comments and Suggestions for Authors

The revied manuscript is improved. However, the authors have not yet addressed all my suggestions.

1.     It is important to examine and show the advantages of the tandem endoscopy, including cost, procedure time, tolerance, and safety (complications) as the next step. I recommend that the authors comment the future prospect n the manuscript.

2.     Results and Discussion seems redundant. Treatment and clinically-unimportant endoscopic findings such as fundic gland polyp should be more summarized, focused on the findings related to premalignant and malignant lesions. In addition, the results regarding stomach can more summarized, without separating each region.
